# Time Series High-Resolution Land Surface Albedo Estimation Based on the Ensemble Kalman Filter Algorithm

**Guodong Zhang [1,2], Hongmin Zhou [1,\*], Changjing Wang [1,2], Huazhu Xue [2,\*], Jindi Wang [1] and Huawei Wan [3]**

[1] State Key Laboratory of Remote Sensing Science, Beijing Engineering Research Center for Global Land Remote Sensing Products, Faculty of Geographical Science, BNU, Beijing 100875, China; 201704020045@home.hpu.edu.cn (G.Z.); 201704020046@home.hpu.edu.cn (C.W.); wangjd@bnu.edu.cn (J.W.)

[2] School of Surveying & Land Information Engineering, Henan Polytechnic University, Henan 454000, China

[3] Satellite Environment Center, Ministry of Environmental Protection, Beijing 100094, China; wanhw@secmep.cn

\* Correspondence: zhouhm@bnu.edu.cn (H.Z.); xhz@hpu.edu.cn (H.X.); Tel.: +86-10-58806011 (H.Z.); +86-391-3987661 (H.X.)

**Abstract:** Continuous, long-term sequence, land surface albedo data have crucial significance for climate simulations and land surface process research. Sensors such as the Moderate-Resolution Imaging Spectroradiometer (MODIS) and Visible Infrared Imaging Radiometer (VIIRS) provide global albedo product data sets with a spatial resolution of 500 m over long time periods. There is demand for new high-resolution albedo data for regional applications. High-resolution observations are often unavailable due to cloud contamination, which makes it difficult to obtain time series albedo estimations. This paper proposes an "amalgamation albedo" approach to generate daily land surface shortwave albedo with 30 m spatial resolution using Landsat data and the MODIS Bidirectional Reflectance Distribution Functions (BRDF)/Albedo product MCD43A3 (V006). Historical MODIS land surface albedo products were averaged to obtain an albedo estimation background, which was used to construct the albedo dynamic model. The Thematic Mapper (TM) albedo derived via direct estimation approach was then introduced to generate high spatial-temporal resolution albedo data based on the Ensemble Kalman Filter algorithm (EnKF). Estimation results were compared to field observations for cropland, deciduous broadleaf forest, evergreen needleleaf forest, grassland, and evergreen broadleaf forest domains. The results indicated that for all land cover types, the estimated albedos coincided with ground measurements at a root mean squared error (RMSE) of 0.0085–0.0152. The proposed algorithm was then applied to regional time series albedo estimation; the results indicated that it captured spatial and temporal variation patterns for each site. Taken together, our results suggest that the amalgamation albedo approach is a feasible solution to generate albedo data sets with high spatio-temporal resolution.

**Keywords:** land surface albedo; time series; high spatio-temporal resolution; EnKF

## 1. Introduction

Land surface albedo, which is defined as the fraction of incident solar radiation (0.3–5.0 μm) reflected by land surfaces [1], is widely used in ground energy balance analysis, weather climate prediction, and climate change research [2,3]. Existing global albedo products include the Advanced Very High-resolution Radiometer (AVHRR) [4–6], Earth Radiation Budget Experiment (ERBE) [7], Moderate-Resolution Imaging Spectroradiometer (MODIS) [8], Multi-angle Imaging

SpectroRadiometer (MISR) [9], POLarization and Directionality of the Earth's Reflectances (POLDER) [10–12], Meteorological Satellites (Meteosat) [13], and Global Land Surface Satellite (GLASS) [14], are all conventional production. Their spatial resolution ranges from several hundred meters to several kilometers, which is much larger than the characteristic patch size for forest management, the typical field size of global agriculture, or the dominant extent of impervious surfaces in urban areas [15]. There is yet demand for new land surface albedo estimation methodologies which span longer time periods without sacrificing high spatio-temporal resolution.

Previous researchers have proposed various techniques for estimating high spatial resolution albedo with satellite imagery. Liang [16], for example, used an extensive radiative transfer model as a conversion formula for calculating total shortwave albedo, total-, direct-, and diffuse-visible, and near-infrared broadband albedos for several narrowband sensors. He [17,18] later applied the algorithm to the Chinese environment and disaster monitoring and forecasting small satellite constellation (HJ) data and Landsat series data to derive 30 m resolution albedo data. The results were validated at Surface Radiation (SURFRAD), AmeriFlux, Baseline Surface Radiation Network (BSRN), and Greenland Climate Network (GC-Net) sites; the direct estimation algorithm did provide accurate albedo estimations for different land cover types with root mean squared errors (RMSEs) ranging from 0.022 to 0.034 for snow-free surfaces. Shuai et al. [19] proposed an algorithm for generating land surface albedo at 30 m resolution using Landsat and the anisotropy information from Moderate-Resolution Imaging Spectroradiometer (MODIS) observations. Their estimated albedos showed an absolute accuracy of $\pm$ 0.02–0.05, RMSE less than 0.03, and a bias less than 0.02 by comparison against field measurements. Zhang [20] used the Spatial and Temporal Adaptive Reflectance Fusion Model (STARFM) to blend spatial information from fine-resolution shortwave albedo images and temporal information from coarse-resolution shortwave albedo images to successfully estimate high spatio-temporal resolution albedos. These methods all center on the use of Landsat imagery to obtain high spatial resolution albedos. However, Landsat data sets are affected by the long satellite return cycle and cloud contamination, and do not readily provide surface albedo data with high temporal resolution or time series albedo data.

Data assimilation is an effective approach to time series land surface parameter estimation. It is a mechanism to integrate various direct or indirect observation information types of varying resolution and from different sources to automatically adjust the model trajectory which provides accurate dynamic model state and model predictions. The Kalman filter algorithm was first developed in 1960 and later used to estimate time series soil moisture, leaf area index (LAI), and other surface parameters [21,22]. Li [23] proposed the Dual ensemble Kalman Filter (Dual EnKF) to estimate a time series LAI; Dual EnKF represents an updated LAI estimation with more sensitive parameters (LAI, Markov parameter, weight of the first price function, and weight of the second price function) in the dynamic model. Zhou [24] built a data-based mechanistic assimilation technique by coupling a revised universal data-based mechanistic model (LAI_UDBM) with a vegetation canopy radiative transfer model (PROSAIL). The Ensemble Kalman filtering algorithm was applied to enhance the LAI estimation accuracy.

Many other researchers have achieved notable results in this field. Shi [25] used the China Land Soil Moisture Date Assimilation System (CLSMDAS) to perform an assimilation experiment on soil moisture based on the EnKF and Land Surface Process Model; their results reasonably reflected the spatial and temporal distribution of soil moisture. Jin [26] used the EnKF algorithm coupled with the canopy radiative transfer model (ACRM) to accurately predict time series LAI data from a phenological model. Xu [27] proposed an algorithm that integrated Direct Insertion (DI) and Deterministic Ensemble Kalman Filter (DEnKF) methods to assimilate snow depth with surface albedo; the solution was shown to improve the precision of snow depth simulations. Although the EnKF assimilation method has been widely used in the inversion of many surface parameters, it has not yet been applied to the estimation of land surface albedos.

In this paper, we propose a time series high-resolution land surface albedo estimation algorithm based on the ensemble Kalman filter. The albedo dynamic model was established based on historical MODIS albedo products and Thematic Mapper (TM) observations, and albedo data was recursively updated via the EnKF method.

## 2. Study Areas and Data

### 2.1. Study Areas

We selected 18 FluxNet sites as our research areas. Among them, 15 sites were used for estimating and verifying the single-point time series albedo for five different land cover types (cropland; deciduous broadleaf forest; evergreen needleleaf forest; grassland; evergreen broadleaf forest). Five sites were selected as the central pixel for estimating and verifying regional time series albedos for each land cover type.

Due to the long satellite reentry cycle and cloud contamination, satellite data and ground data did not always match temporally. We also needed to match ground data with TM data in the time series to conduct this study, so the year of data used for each site in the experiment was inconsistent . Table 1 provides further information.

**Table 1.** Ground stations used for validation.

| Site Name | Network | Latitude (°) | Longitude (°) | Land Cover Type | Data Year |
|-----------|---------|--------------|---------------|-----------------|-----------|
| CA-Oas * | FluxNet | 53.6289N | 106.1978W | DBF | 2009 |
| IT-Col * | FluxNet | 41.8494N | 13.5881E | DBF | 2009 |
| IT-Ro2 * | FluxNet | 42.3903N | 11.9209E | DBF | 2010 |
| US-Bar + | AmeriFlux | 44.0646N | 71.2881W | DBF | 2009 |
| FR-Pue * | FluxNet | 43.7414N | 3.5958E | EBF | 2009 |
| MY-Pso * | FluxNet | 2.9730N | 102.3062E | EBF | 2009 |
| AU-Wac *+ | FluxNet | 37.4259S | 145.1878E | EBF | 2007 |
| CH-Dav * | FluxNet | 46.8153N | 9.8559E | ENF | 2009 |
| FI-Hyy * | FluxNet | 61.8474N | 24.2948E | ENF | 2009 |
| NL-Loo * | FluxNet | 52.1666N | 5.7436E | ENF | 2010 |
| IT-Sro + | FluxNet | 43.7279N | 10.2844E | ENF | 2009 |
| DE-Kli * | FluxNet | 50.8931N | 13.5224E | CRO | 2009 |
| IT-Bci * | FluxNet | 40.5238N | 14.9574E | CRO | 2010 |
| FR-Gri * | FluxNet | 48.8442N | 1.9519E | CRO | 2010 |
| US-Arm + | AmeriFlux | 36.6058N | 97.4888W | CRO | 2010 |
| AU-DaP * | FluxNet | 14.0633S | 131.3181E | GRA | 2008 |
| US-Ib2 * | AmeriFlux | 41.8406N | 88.2410W | GRA | 2010 |
| AU-Stp *+ | FluxNet | 17.1507S | 133.3502E | GRA | 2009 |

Cropland (CRO); deciduous broadleaf forest (DBF); evergreen needleleaf forest (ENF); grassland (GRA); evergreen broadleaf forest (EBF). Sites with * are used for estimating and verifying single-point time series albedo, sites with + are selected as the central pixel for estimating and verifying regional time series albedo.

### 2.2. Ground Verification Data

FluxNet is a global network of micrometeorological flux measurement sites that measure the exchanges of carbon dioxide, water vapor, and energy between the biosphere and atmosphere. It was established based on other observation networks including AmeriFlux, CarboEurope, AsiaFlux, OzFlux, and a few independent sites. At present, over 140 Flux tower stations are operating on a long-term and continuous basis. Data and site information are available online at the FluxNet website, http://fluxnet.fluxdata.org/. Land surface types include temperate conifer and broadleaf (deciduous and evergreen) forests, tropical and boreal forests, crops, grasslands, chaparral, wetlands, and tundra. Sites exist on five continents and their latitudinal distribution ranges from 70 °N to 30 °S. The FluxNet site records upwelling and downwelling shortwave radiative flux data with a half-hour observation period. For our purposes, we selected observation data from 0.5 hours before

and after noon. The shortwave albedo of the site was calculated as the ratio of upwelling radiation and downwelling radiation. After eliminating invalid observations (filled data) and data with albedos less than 0 or greater than 1, each selected site was associated with sufficient time series ground albedo data for validation.

### 2.3. Landsat Satellite Data

Landsat sensors have continuously imaged the land surface since the 1970s [28]. The TM onboard Landsat 4 and 5 satellites with seven spectral bands sampled the shortwave range at a spatial resolution of 30 m from 1984 to 2011. TM images are an important remote sensing data source for earth resources and environments as per their high spatial resolution, spectral resolution, and positioning accuracy. We downloaded all available Landsat data during the ground measurement period. The distribution of available high-quality TM data corresponding to each FluxNet site within one year is shown in Figure 1. Abundant cloud-free TM data can be obtained for individual pixels; TM data are prone to contamination over larger areas. As shown in Figure 1, all of the sites have more than five available cloud-free TM images within one year. This provides reliable observation data for the effective operation of the EnKF algorithm.

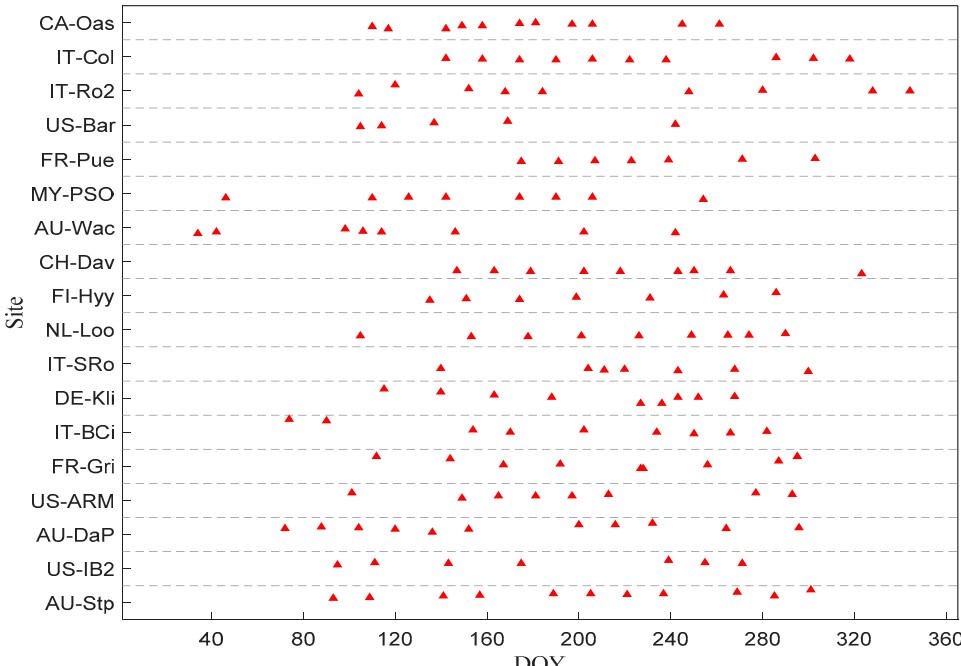

**Figure 1.** Distribution of available Thematic Mapper (TM) data corresponding to Flux sites (one year).

### 2.4. MCD43A3 BRDF/Albedo Product

MCD43A3 (Version 6) (V006) is the latest version of the MODIS Bidirectional Reflectance Distribution Functions (BRDF)/Albedo product. It includes bi-hemispherical reflectance (white-sky albedo) and directional-hemispherical reflectance (black-sky albedo) with a 500 m spatial resolution. The daily albedo is composed of 16-day multi-angle observations, where the Julian date of each specific file represents the 9th day of the 16-day retrieval period. MCD43A3 was produced based on the Algorithm for Model Bidirectional Reflectance Anisotropies of the Land Surface (AMBRALS), which uses all atmospherically corrected, high-quality, cloudless surface reflectance over the course of 16 days to achieve the best-fit surface bidirectional reflectivity via Ross–Li kernel models [29,30]. The Ross–Li kernel model is expressed as follows:

$$R(\theta, \vartheta, \phi, \Lambda) = f_{iso}(\Lambda) + f_{vol}(\Lambda)K_{vol}(\theta, \vartheta, \phi) + f_{geo}(\Lambda)K_{geo}(\theta, \vartheta, \phi) \tag{1}$$

where $R(\theta, \vartheta, \phi, \Lambda)$ is the frontal reflectivity with a solar zenith angle $\theta$, observed zenith angle $\vartheta$, relative azimuth $\phi$, and wavelength band $\Lambda$. $f_{iso}(\Lambda)$ is the proportion of uniform scattering in all directions, $f_{vol}(\Lambda)$ is the proportion of body scattering, $f_{geo}(\Lambda)$ is the proportion of geometric optical scattering, $K_{vol}(\theta, \vartheta, \phi)$ is the RossThick kernel, and $K_{geo}(\theta, \vartheta, \phi)$ is the LiSparse kernel.

MODIS V006 products provide shortwave black-sky albedo and white-sky albedo [31], which must be converted into blue-sky shortwave albedo according to the proportion of sky scattered light [32].

$$\alpha(\theta_i, \lambda) = (1 - s(\theta_i \tau(\lambda)))\alpha_{bs}(\theta_i, \lambda) + s(\theta_i \tau(\lambda))\alpha_{ws}(\theta_i, \lambda) \tag{2}$$

where $\alpha(\theta_i, \lambda)$ is the blue-sky albedo of the band $\lambda$ at a solar zenith angle of $\theta$, $\alpha_{bs}(\theta_i, \lambda)$ is the black-sky albedo, $\alpha_{ws}(\theta_i, \lambda)$ is the white-sky albedo, and $s(\theta_i, \tau(\lambda))$ is the fraction of diffuse skylight when the solar zenith angle is $\theta$, which is a function of aerosol optical depth and can be calculated using a predetermined look-up table (LUT) based on the 6S atmospheric radiative transfer code [33].

There was some data missing in MCD43A3 due to retrieval failure, so we used a gap filling algorithm to construct a continuous albedo data set. Missing data was filled with the average of all pixel values of the same land type in a given window ($10 \times 10$ pixels) centered on the target pixel.

## 3. Methods

Our data assimilation methodology included an EnKF which recursively updated the albedo by coupling the direct estimation approach with a dynamic model. A flow chart of this process is shown in Figure 2. The existing high quality multi-year MODIS albedo product data is averaged to compute albedo climatology, then a simple dynamic model is established based on the climatology to evolve albedo over time and to forecast short-range albedos. A direct estimation approach is used to generate high-resolution TM albedo data. The EnKF technique is used to estimate real-time albedos by combining the predictions from the dynamic model and the high-resolution TM data.

The proposed method was essentially a three-step process:

(1) Obtain MODIS albedo data for the historical period of interest, correct them geometrically, convert black-sky albedo (BSA) and white-sky albedo (WSA) to blue-sky albedo using the scatter ratio, resample to a 30 m resolution, and average the albedo of the historical period to obtain the time-series shortwave albedo climatology;

(2) Transform the cloudless TM image into a 30 m spatial resolution TM shortwave albedo via direct estimation algorithm; and

(3) Use the EnKF method with the MODIS time series albedo as the background field and input the TM high spatial resolution albedo data to estimate the high spatial resolution albedo.

This method effectively integrated the time change information from MODIS data and spatial Landsat data to resolve issues with the low spatial resolution, sparsity, and long return cycle of MODIS data, as well as the sparsity of Landsat data to ultimately obtain high spatial and temporal resolution albedo data.

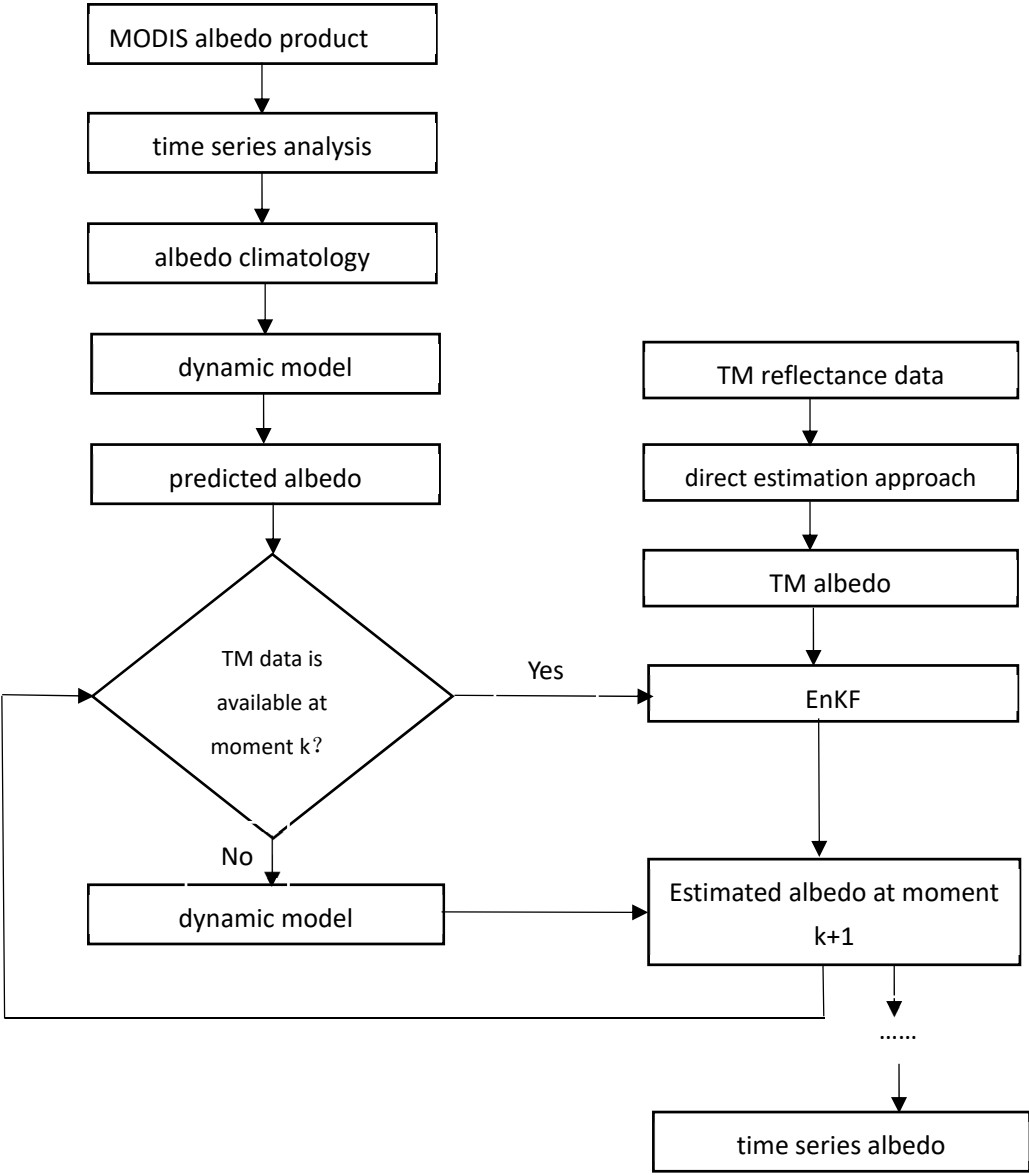

**Figure 2.** Real-time albedo inversion based on ensemble Kalman Filter (EnKF).

### 3.1. Validation Site Land Surface Heterogeneity

Previous researchers have verified the consistency of global land surface albedo products against ground measurement data at MODIS 500 m spatial scales at SURFRAD, Atmospheric Radiation Measurement Southern Great Plains (ARM/SGP) [34], and Cloud and Radiation Testbed-Southern Great Plains (CART/SGP) sites [35]. The MODIS albedo product has high accuracy with RMSE of 0.013–0.018. However, these verifications are based on homogeneous sites using point observations. In heterogeneous surface areas, site observations do not represent the MODIS pixels area, and thus the remote sensing products cannot be fully verified. As such, we must analyze site representativeness before validation with remote sensing data [36]. The heterogeneity of surface space describes the extent of surface heterogeneity within a certain range and reflects the heterogeneity in the area around FluxNet sites [37,38]. In this study, we used variogram model parameters and the relative coefficient of variation ($R_{CV}$) to analyze the effect of heterogeneity in the area around FluxNet sites according to Roman [39].

The isotropic spherical variogram model was used to evaluate landscape heterogeneity with 1.0 $km^2$, 1.5 $km^2$, and 2.0 $km^2$ TM albedo subsets. The TM albedo was obtained using the direct estimation approach proposed by He [18]. The spherical model formula is:

$$\gamma_{sph}(h) = \begin{cases} c_0 + c \cdot \left(1.5 \cdot \frac{h}{a} - 0.5(\frac{h}{a})^3\right) & \text{for } 0 \le h \le a \\ c_0 + c & \text{for } h > a \end{cases} \tag{3}$$

The range a defines the distance from a point beyond which there is no further correlation of a biophysical property associated with that point. It has also been described as the average patch size of the landscape when the curve reaches the high-level distance [40]. The data are correlated within the range of a. Otherwise, the data are not correlated with each other; that is, the observations outside the range do not affect the estimation results. For a 1.0 $km^2$ subset $a_{max}$ = 690 m, for a 1.5 $km^2$ subset $a_{max}$ = 1050 m, and for a 2.0 $km^2$ subset $a_{max}$ = 1410 m. The sill c is the variation of ordinates when the abscissa is greater than the variable range. The nugget $c_0$ is the variation of the abscissa at 0, which describes the variability of the data at the microcosmic level.

The coefficient of variation (CV) is the ratio of the standard deviation to the mean. It is independent of spatial scale and can provide an estimate of the overall variability of data. The relative CV is:

$$R_{cv} = \frac{CV_{1.5x} - CV_x}{CV_x}; x = 1.0 \text{ km}^2 \tag{4}$$

where $R_{cv}$ is the relative CV, $CV_{1.5x}$ is the CV calculated from a 1.5 $km^2$ TM albedo subset at the center of a given observation station, and $CV_x$ is the CV calculated from a 1.0 $km^2$ TM albedo subset at the center of the given observation station. If a site is more representative, it has a weaker spatial heterogeneity, a similar landscape around the site, and $R_{cv}$ closer to 0.

*3.2. Albedo Dynamic Model Construction*

Albedo climatology is determined by the average of albedo over historical period. In this study, we calculated albedo climatology from high-quality multi-year MODIS albedo data as follows:

$$ALB_t' = \frac{1}{n}\sum_{k=1}^{n} ALB_t(k) \tag{5}$$

where $ALB_t'$ is the albedo value corresponding to time t on the background field curve, $ALB_t(k)$ is the MODIS albedo value at time t, and n is the year.

The albedo climatology describes the general change tendency of albedo in a one-year period. There may be some deviation from the ground measurements due to precipitation, anthropogenic activities, and other reasons. Albedo climatology simulates the general trend of land surface albedo and supplies the albedo estimation background. When new observations are available, the EnKF updates the estimation to produce a final value. We constructed our dynamic model based on the climatology used to forecast the short-range albedo:

$$ALB_t = F_t \times ALB_{t-1} \tag{6}$$

where $ALB_t$ represents the current estimated albedo, $ALB_{t-1}$ represents the estimated albedo at the preceding time step, and $F_t$ is:

$$F_t = 1 + \frac{1}{ALB_t + \varepsilon} \times \frac{dALB_t}{dt} \tag{7}$$

where $\varepsilon = 10^{-3}$ prevents negative denominators and $\frac{dALB_t}{dt}$ is the growth rate of albedo at time t. The dynamic model was adopted from Samain et al. [41] and Xiao et al. [42].

### 3.3. High-Resolution Albedo Estimation

Surface albedo can be obtained directly from Top of Atmosphere (TOA) reflectance without requiring atmospheric correction [43]. The apparent surface albedo was separated from the inherent surface albedo, and the empirical relationship between the simulated surface albedo and TOA reflectance was built based on extensive radiative transfer simulations under a variety of atmospheric conditions. To mitigate errors from nonlinearities, a new statistical relationship was extended to generate albedo measurements from the spectral bands and three broadbands [18], including the visible (300–700 nm), near-infrared (NIR; 700–3000 nm), and total shortwave ranges (300–3000 nm), based on the following linear equation:

$$\alpha_\lambda = \sum \rho_i^{TOA} \cdot c_i + c_0 \tag{8}$$

where $\alpha_\lambda$ is the surface albedo for the spectral range of $\lambda$, $\rho_i^{TOA}$ is the TOA reflectance for spectral band i, and $c_i$ and $c_0$ are the regression coefficients.

The direct estimation approach simulates TOA reflectance from radiative transfer information. Here, the sensor spectral response functions obtained from the USGS and the MODIS BRDF database were used as inputs for the 6S radiative transfer code, aerosol types, water vapor, ozone, and $CO_2$. The linear regression coefficients for each of the geometrical combinations were precalculated and stored in LUTs for operational use. TOA reflectance was simulated using the regression models in the LUT. Cloud-free observations were selected from the quality control document, then the TM surface albedo was obtained from TOA reflectance data. The direct estimation approach has been successfully applied to Landsat Multispectral Scanner (MSS), Thematic Mapper (TM), Enhanced Thematic Mapper Plus (ETM+), and Operational Land Imager (OLI) sensors [18]; extensive validations have also been carried out at SURFRAD, AmeriFlux, BSRN, and GC-Net sites. The direct estimation approach can generate reliable surface albedo estimates with accuracy of 0.022 to 0.034 in terms of RMSEs over snow-free surfaces [44,45]. The direct estimation approach has also been used in OLI and Gaofen-1 satellite (GF-1) surface albedo inversion [46,47].

### 3.4. Ensemble Kalman Filter

In the 1960s, Kalman [48,49] and Bucy [48] developed the Kalman filter algorithm for optimal estimation of system states according to a linear system state equation and combined input and output system observation data. The traditional Kalman filter algorithm is often applied to linear problems [50,51]. It can estimate the state of a dynamic system from a series of data with measurement noise when the measurement variance is known, and is a common component in communication, navigation, guidance, and control applications. The dynamic model and observer in a Kalman filter are typically expressed as follows:

$$x_{k+1} = M_k x_k + \omega_{k+1} \tag{9}$$

$$z_k = H_k x_k + v_k \tag{10}$$

where $x_k$ is the dynamic model state vector; $M_k \in R^{n \times n}$ is the system matrix, which changes with time; $z_k$ represents observed values, and $H_k$ is a time-varying observation system which transforms the state variable into the same value as the observed value. In this study, the observed values and state variable were land surface albedos. $\omega_k$ is the model error and $v_k$ is observed noise. The albedo has different levels of inversion accuracy over time.

The predicted value of the state variable of the dynamic model at time k is defined as $x_k^f$. The data assimilation step serves to calculate the optimal estimate $x_k^a$ of the state variables at moment k by combining the predicted value $x_k^f$ and the observed value $z_k$ of the model state variables at time k, $x_k^a$, the optimal estimate of $x_k$, is a linear function of $x_k^f$ and $z_k$.

$$x_k^a = x_k^f + K_k [z_k - H_k x_k^f] \tag{11}$$

where $z_k - H_k x_k^f$ is observed incrementally (innovation) and $K_k$ is the Kalman gain matrix. Data assimilation is conducted to minimize the variance of $x_k^a$ by determining $K_k$. The update equations for state variables and covariance are:

$$x_k^a = x_k^f + K_k[z_k - H_k x_k^f] = x_k^f + P_k^f H_k^T (H_k P_k^f H_k^T + R_k)^{-1}(z_k - H_k x_k^f) \tag{12}$$

$$P_k^a = P_k^f - K_k H_k P_k^f = P_k^f - P_k^f H_k^T (H_k P_k^f H_k^T + R_k)^{-1} H_k P_k^f \tag{13}$$

where $x_k^a$ is the updated state variable value, $x_k^f$ is the predicted state variable, $P_k^f$ is the predicted covariance, and $P_k^{pa}$ is the posterior covariance of state variables.

The ensemble Kalman filter algorithm is a complex sequential data assimilation method which produces non-linear models within a Kalman gain scheme. In the ensemble Kalman filter algorithm, x is defined as an n-dimensional model state vector and $A = (x_1, x_2, \ldots x_N) \in \Re^{n \times N}$ is composed of a collection of N model state vectors. The ensemble mean is stored in each column of $\overline{A} \in \Re^{n \times N}$. The ensemble perturbation matrix is defined as:

$$A' = A - \overline{A} \tag{14}$$

The ensemble covariance matrix is:

$$P_e = \frac{A'(A')^T}{N - 1} \tag{15}$$

Given a vector of measurements $d \in \Re^m$, where m is the number of measurements, the observation matrix is expressed as follows:

$$D = (d_1, d_2, \ldots d_N) \in \Re^{m \times N} \tag{16}$$

Then, the standard analysis equation of the Kalman filter is:

$$A^a = A + A' A'^T H^T (H A' A'^T H + R)^{-1}(D - HA) \tag{17}$$

where $H \in \Re^{n \times N}$ (m is the number of measurements) is the measurement operator relating the model state to the observations; $R \in \Re^{m \times m}$ is the observational error covariance matrix, and $D \in \Re^{m \times N}$ is the disturbance observation matrix set. H is a linear operator which is not suitable in cases when the operator is nonlinear. By augmenting the model state vector, $\overline{x}^T = [x^T, h^T(x)]$. $\overline{Y} \in \Re^{n \times N}$ is defined as the set matrix of augmented state vectors and $\overline{Y'} \in \Re^{n \times N}$ is the set disturbance matrix of state variables. As such, the analytical equation is:

$$A^a = A + A' \overline{Y'}^T \overline{H'}^T (\overline{HY'Y'}^T \overline{H'}^T + R)^{-1}(D - \overline{HY}) \tag{18}$$

where $\overline{H}$ is a new observation operator. The analytical equation can solve the data assimilation problem for which the observation operator is nonlinear.

In this study, we calculated the background error covariance from albedo background and field observations. We calculated observation error covariance values from TM albedo and field observations over the time series described above. At each time point, we generated N random noises with 0 as the mean and 0.01 as the variance. N is the ensemble number, which affects estimation efficiency and accuracy: A larger N has a lower calculation efficiency and higher accuracy. We set the ensemble number to 100 and added random noise values to the background field values of the current date to obtain N new background field values [52]. Finally, we calculated the standard deviation between the N background field values and the ground field observations as the background field errors of the current date. Observation errors were simulated in the same way. Figure 3 shows the error setting for the background field, TM albedo, and field observation for the AU-DaP site in 2008.

The "background error" and "TM error" labels represent simulated background errors and TM albedo errors, respectively. For regional use, the error of the area was set based on the error of the center pixel; a normal distribution noise with a mean of 0 and a variance of 0.01 was added for each pixel.

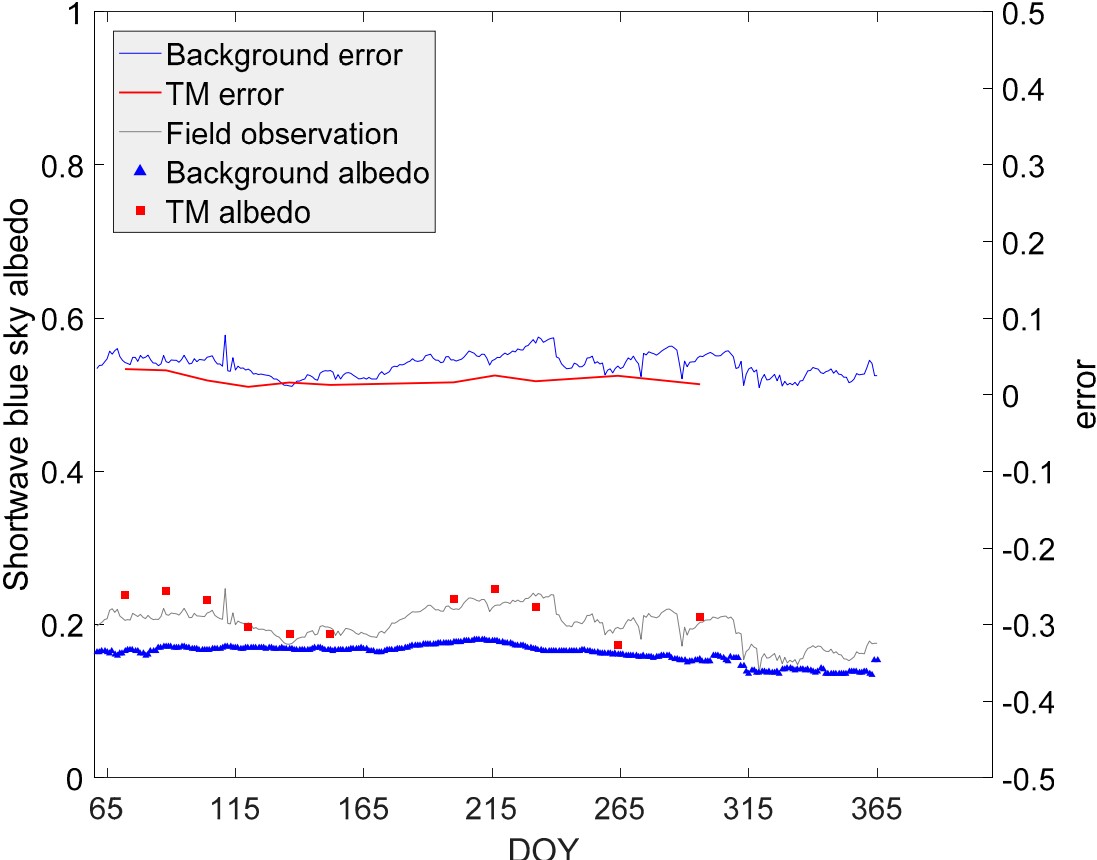

**Figure 3.** Error setting for albedo background and TM data.

Surface albedo is the only model state here, so the size of the model state vector n is equal to 1. We used the albedo value derived by the direct estimation approach as the albedo observation data, so the sum of the size of the model state vector n and the number of measurement equivalents added to the original model state vector was equal to 2. The simulated albedo was included in the state variables of the dynamic model at the given observation time. We emphasize the observation error covariance R here as per its significant influence on the $A^a$ value. In our experiments, all the errors were simulated at site AU-DaP.

*3.5. Validation*

We used the time series field-observed local noon albedos at FluxNet sites to validate the albedos estimated with the EnKF algorithm. Spatially continuous albedos derived by the direct estimation approach were selected to validate the accuracy of estimated albedo at the regional scale. Single-point verification was performed at 15 FluxNet sites as shown in Table 1; the spatial representativeness of each FluxNet site was investigated according to the method described in Section 3.1. Three statistical parameters, coefficient of determination ($R^2$), RMSE and Bias, were used to evaluate the results. If the maximum error between the estimated result and the validated value did not exceed 0.05, the estimated result was considered accurate enough for scientific use [1,53].

## 4. Results

### 4.1. Land Surface Heterogeneity at FluxNet Sites

Figure 4a shows an 18 km subset TM albedo for IT-Col and a magnified image of the measurement point and spatial boundaries of 1.0 km$^2$ and 1.5 km$^2$, which provides a detailed visual representation of the landscape heterogeneity at the site. Figure 4b shows the variogram estimator (point values) for the TM albedo subsets. The spatial variability is more obvious over the larger squared regions (1.5 km$^2$ and 2.0 km$^2$), and different land cover types are more likely to be found at larger separation distances. This variogram estimator reaches an asymptote or constant variance between spatial uncorrelated samples near the sample variance. We calculated the range and $R_{cv}$ for each Flux site to determine the heterogeneity of MODIS pixels as summarized in Table 2.

When the TM footprint increased from 1.0 km to 1.5 km, $R_{cv}$ increased 0%–10% over CA-Oas, IT-Col, IT-Ro2, FR-Pue, AU-Wac, CH-Dav, NL-Loo, DE-Kli, IT-BCi, FR-Gri, US-ARM, AU-DaP, and US-IB2. These sites differ less from the surrounding landscape; when the TM footprint changes, $R_{cv}$ only changes slightly relative to the smaller landscapes in the surrounding regions. The size of $R_{cv}$ depends on the degree of landscape similarity within the given range. When surface heterogeneity around the site is weak, the overall degree of stationarity between regions is similar, the overall variability is small, and the value of $R_{cv}$ is relatively low. $R_{cv}$ increased 32.46% over HI-Fyy and 86.38% over IT-Sro as the TM footprint increased from 1.0 km to 1.5 km. Based on the range of FI-Hyy beyond the 1 km$^2$ limit, the landscapes around the two sites are obviously different; the small lakes near FI-Hyy are not within the 1 km$^2$ boundary, IT-Sro is near the sea, and internal ($CV_{1\,km}$) regions do not include the sea. At the US-Bar, MY-Pso, and AU-Stp stations, the results for $R_{CV}$ were moderate (10%–20%). There are two types of vegetation around the three sites which include mixtures of grass and trees. There are some grasslands located approximately 1.0 km north of the US-Bar tower and southeast of the MY-Pso tower. A broadleaf forest is located southwest of the AU-Stp station, but the internal ($CV_{1\,km}$) does not include the forest. The overall fluctuation between regions was not high—the three sites were moderately heterogeneous.

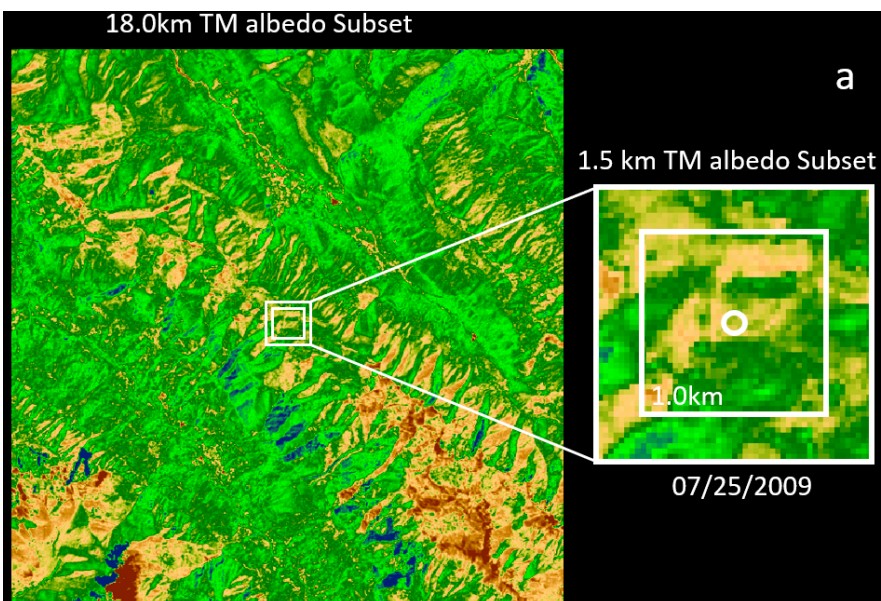

**Figure 4.** *Cont.*

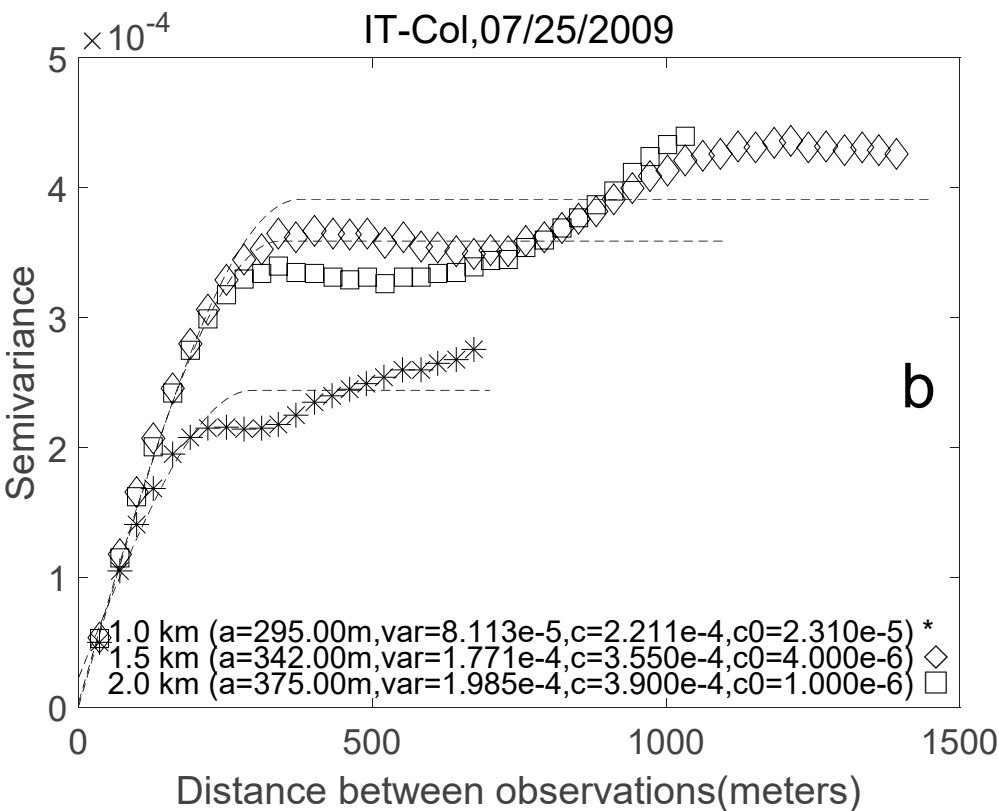

**Figure 4.** (**a**) Subset of 18.0 km TM albedo at IT-Col (measurement point) and spatial boundaries of 1.0 km² and 1.5 km². (**b**) Variogram plot, spherical model (dotted curves), and sample variance obtained via TM albedo on 25 July, 2009 for 1.0 km² (asterisks), 1.5 km² (diamonds), and 2.0 km² (squares) regions.

**Table 2.** $R_{cv}$ for selected ground sites.

| Site | TM Overpass Time | 1 km Range | 1.5 km Range | $R_{cv}$ |
|------|-----------------|-----------|--------------|----------|
| CA-Oas | 10-Aug-09 | 71.00 m | 55.00 m | 5.26% |
| IT-Col | 25-Jul-09 | 295.00 m | 342.00 m | 7.44% |
| IT-Ro2 | 03-Jul-09 | 260.00 m | 129.00 m | 5.38% |
| US-Bar | 17-May-10 | 168.00 m | 353.00 m | 16.87% |
| FR-Pue | 26-Jul-09 | 624.00 m | 332.00 m | 7.8% |
| MY-PSO | 11-Sep-09 | 45.00 m | 644.00 m | 14.38% |
| AU-Wac | 16-Apr-07 | 109.00 m | 112.00 m | 3.40% |
| CH-Dav | 06-Aug-09 | 1453.00 m | 396.00 m | 9.11% |
| FI-Hyy | 31-May-09 | 877.00 m | 155.00 m | 32.46% |
| NL-Loo | 06-Sep-10 | 70.00 m | 34.00 m | 5.33% |
| IT-SRo | 23-Jul-09 | 304.00 m | 641.00 m | 86.38% |
| DE-Kli | 24-Aug-09 | 304.00 m | 193.00 m | 7.25% |
| IT-BCi | 09-Oct-10 | 234.00 m | 182.00 m | 8.94% |
| FR-Gri | 24-May-10 | 114.00 m | 87.00 m | 7.44% |
| US-ARM | 16-Jul-11 | 584.00 m | 393.00 m | 5.53% |
| AU-DaP | 23-Aug-09 | 257.00 m | 422.00 m | 6.74% |
| US-IB2 | 12-Sep-10 | 216.00 m | 243.00 m | 4.30% |
| AU-Stp | 25-Aug-09 | 470.00 m | 341.00 m | 16.77% |

### 4.2. Estimating and Verifying Single-Point Time Series Albedo

We included five vegetated land surface types in this study: Deciduous broadleaf forest, evergreen broad-leaved forest, evergreen coniferous forest, grassland, and farmland. Three sites were subjected to validation for each land surface type. The results are shown in Figure 5.

The algorithm starts with cloudless TM albedo data spanning the whole year. The starting days for each site are different, so the MODIS albedo data for different land surface types have different degrees of overestimation and underestimation; however, they are relatively continuous in time and all reflect the variation characteristics of surface albedo in snow-free periods. The EnKF algorithm combines the advantages of MODIS and TM sensors. The estimated results were very close to the directly calculated TM albedo data when adequate TM data were included. When TM data is absent, the albedo background is used as the final estimation to maintain a complete time series. Our results also showed that with the continuous introduction of TM observations, the errors induced by MODIS background data could be corrected by EnKF. The albedo background is generated by resampling MODIS albedo data.

We also evaluated the heterogeneity of the test sites (Section 4.1) within the entire MODIS 500 m spatial scale. The results indicated that the proposed algorithm can adapt to heterogeneous surfaces (Figure 5e,h,o).

Figure 6 shows scatter plots of the estimated albedos and ground measurements. Different colors represent the three different sites for each land surface type.

As shown in Figure 6, RMSEs of the estimated albedos in deciduous broadleaf forest (DBF), evergreen broadleaf forest (EBF), evergreen needleleaf forest (ENF), cropland (CRO), and grassland (GRA) were 0.0152, 0.0085, 0.0087, 0.0152, and 0.0109; the $R^2$ values were 0.5980, 0.7061, 0.9190, 0.8191, and 0.6911, respectively. The RMSEs for all land surface types were less than 0.02, which meets the requirements of global and regional climate models [53]. The correlation coefficients between the estimated surface albedos and the surface measurements were high ($R^2 > 0.6$). The estimation results were close to the line $y = x$. The accuracies of ENF and CRO were the highest among all land surface types with $R^2$ values of 0.9190 and 0.8191, respectively. The lowest $R^2$ value was 0.5980 for DBF, which can be attributed to the dispersion of TM data (Figure 1). At ENF and CRO sites, the distribution of TM data was more uniform across the whole time series. At the DBF site, there was a slightly longer period of missing data which prevented the albedo from being updated in a timely manner. The estimation accuracy depended on the temporal distribution of the available TM data at a certain degree, as temporally uniform distributed TM data led to higher estimation accuracy. Nevertheless, the proposed algorithm greatly improves the estimation accuracy compared to field observation alone.

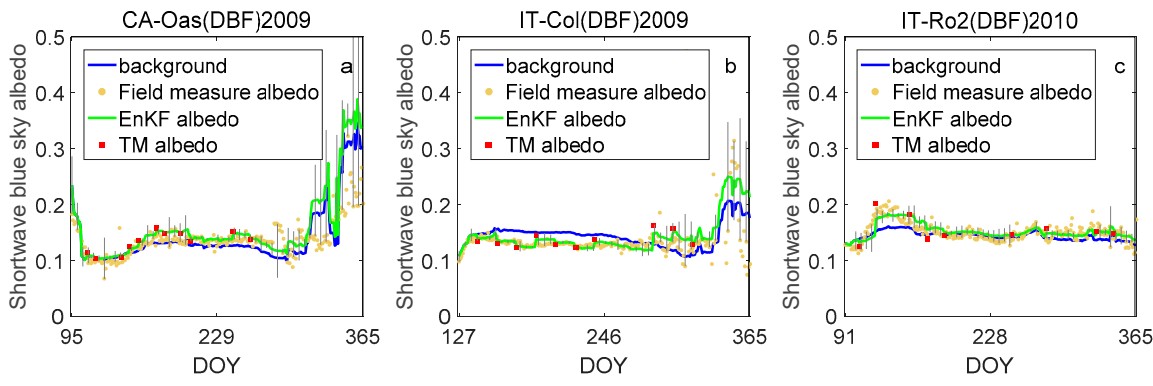

**Figure 5.** *Cont.*

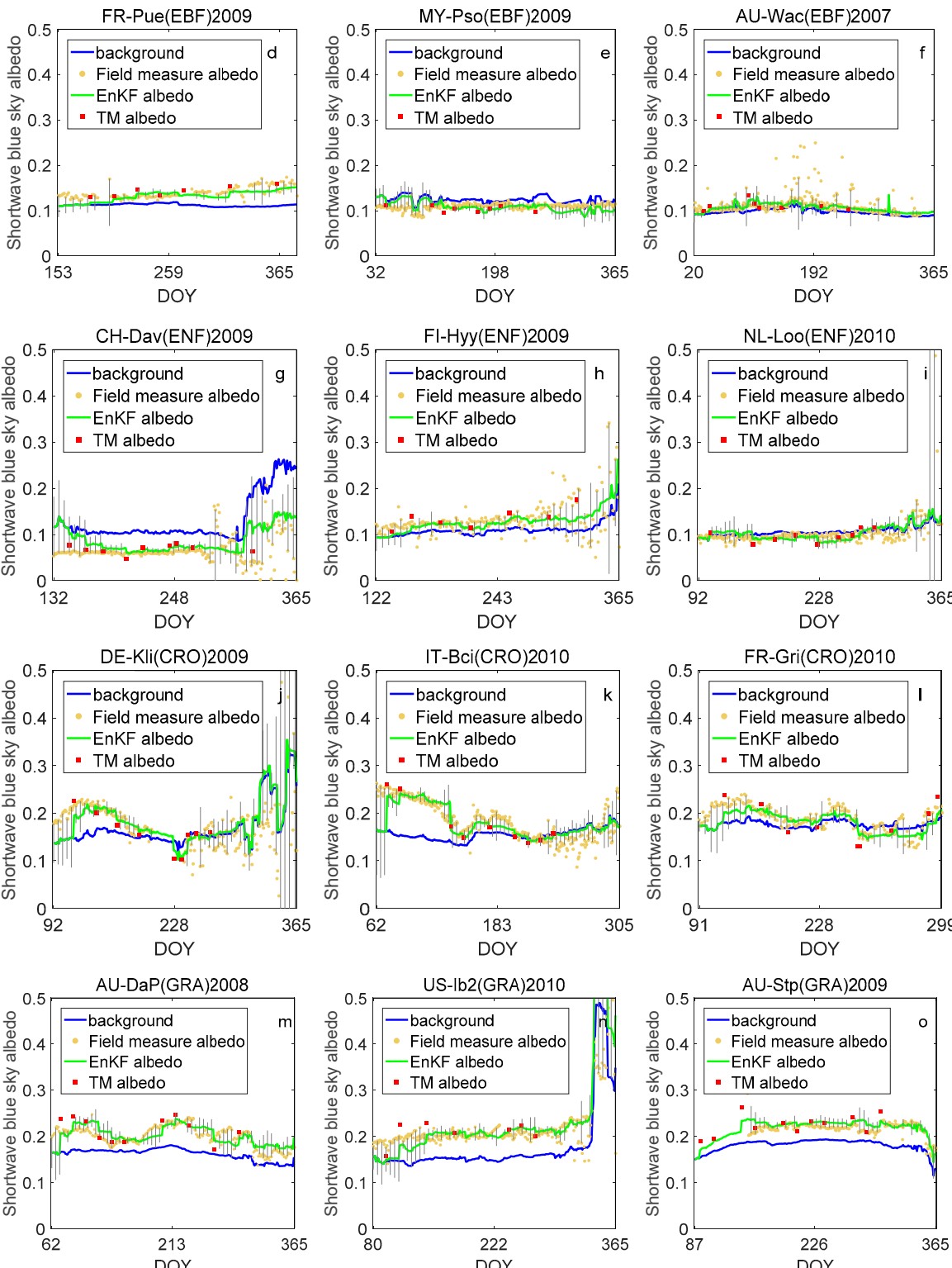

**Figure 5.** Time series albedo estimations based on EnKF for DBF (**a–c**), EBF (**d–f**), ENF (**g–i**), CRO (**j–l**), GRA (**m–o**); three sites for each surface type. Grey vertical line represents error between estimated and measured values.

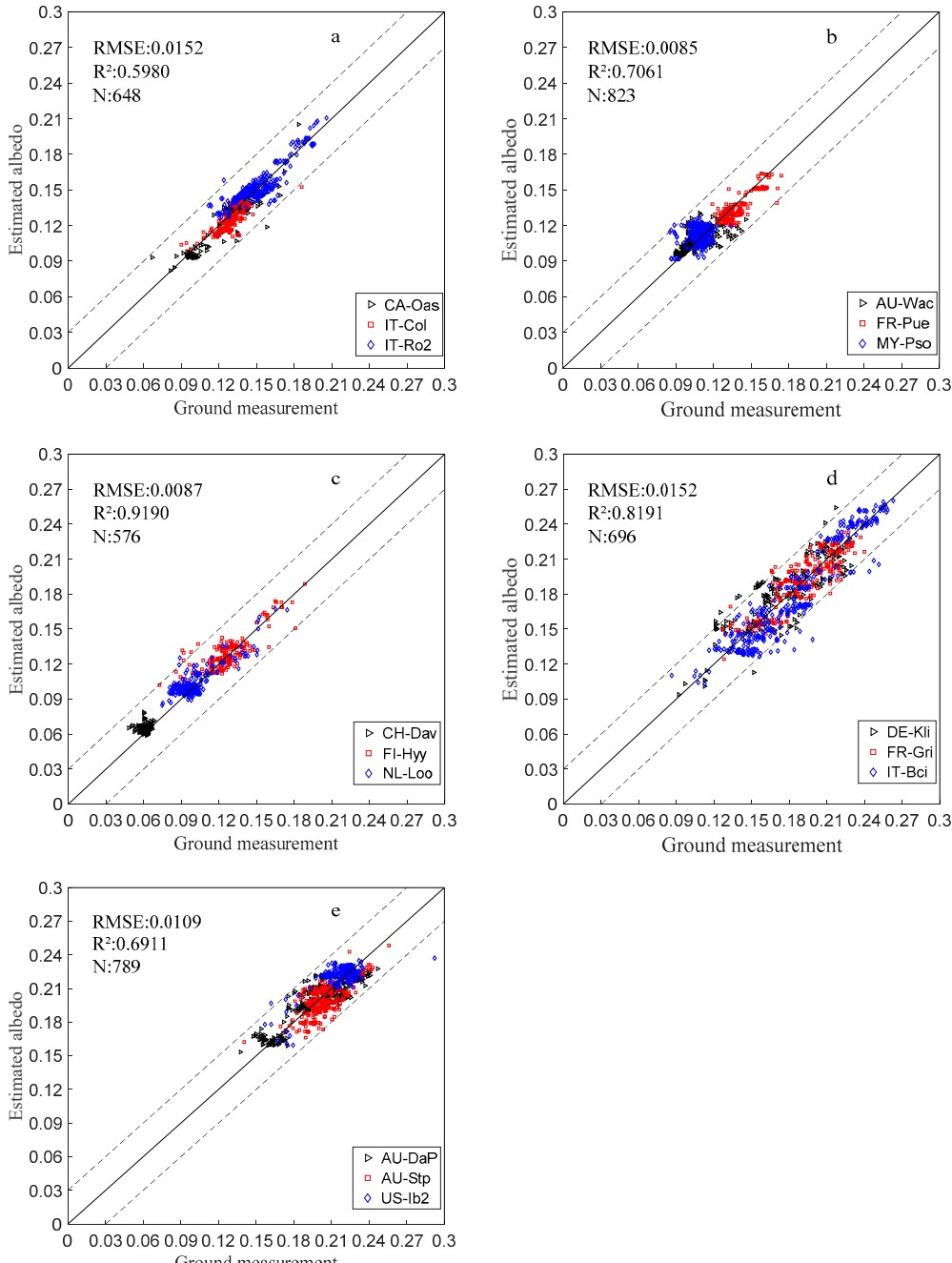

**Figure 6.** Estimated albedos and ground measurements at AmeriFlux and Flxunet sites: (**a**) DBF; (**b**) EBF; (**c**) ENF; (**d**) CRO; (**e**) GRA. Different colors represent different sites.

### 4.3. Regional Timing Albedo Estimation and Verification

We validated the proposed assimilation algorithm on five types of land surface areas, each area containing a ground-based station. The five stations, as mentioned above, were US-Bar, AU-Wac, IT-Sro, US-Arm, and AU-Stp (Table 1). The results are shown in Figure 7. The daily estimation result was difficult to display and the beginning date for each site was different, so we used nine days-worth of estimation results from each site.

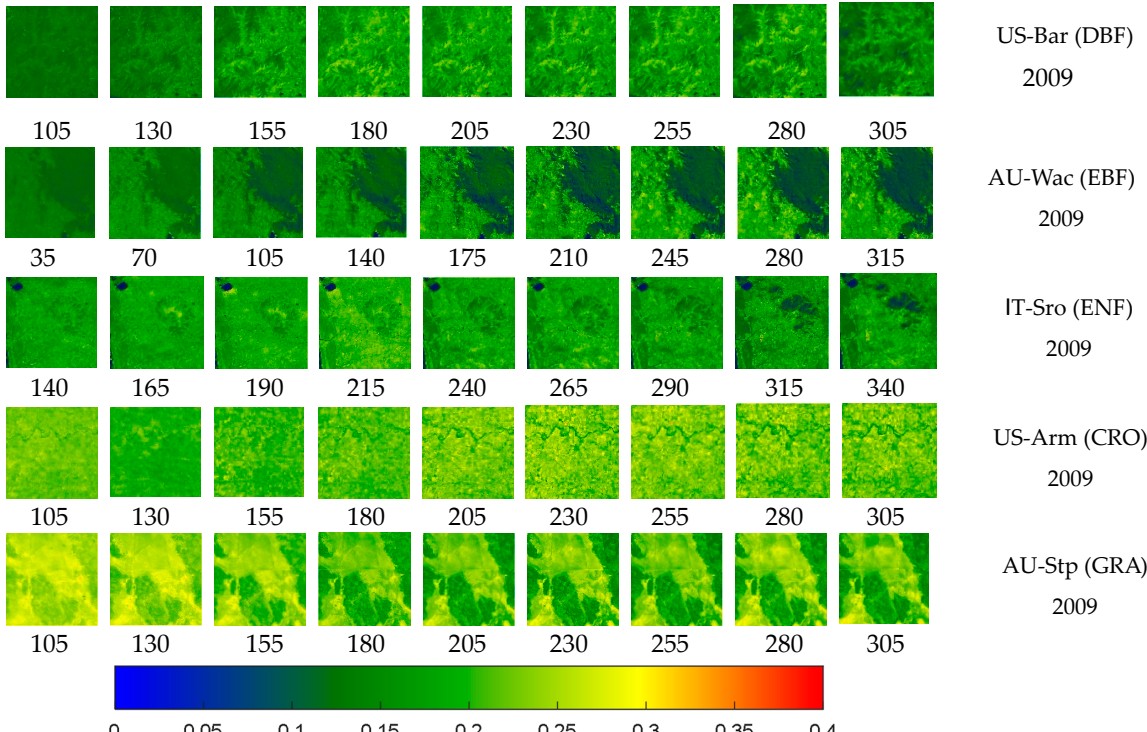

**Figure 7.** Shortwave albedo maps generated by EnKF for five different land surface types. Day of year (DOY) indicated for each albedo map. Estimations have one-day temporal resolution starting with the first cloud-free TM image (only a portion of the results are shown).

The EnKF assimilation method uses TM albedo as observation data. The estimation is strongly dependent on the quantity of TM albedo values. To obtain time-continuous, high spatial resolution albedo data, the EnKF algorithm updates the background field from the first cloudless TM image; therefore, assimilation results will be improved as more TM images become available. Figure 7 shows the estimated time series albedos for five different land cover types. Again, the starting point of each time series was not exactly the same as was determined by the availability of high-quality TM data. Starting with the first cloudless TM image, the algorithm can obtain an albedo dataset with a temporal resolution of one day and a spatial resolution of 30 m (shown here as 25-day or 35-day intervals), thereby reflecting albedo fluctuations within a one-year period.

As verification, we compared the albedo generated by the EnKF algorithm and the TM albedo directly estimated on the same date. Figure 8 shows that the EnKF and TM albedo results are consistent. The scatter plot has RMSE values between 0.0031 and 0.0112, and $R^2$ values between 0.8584 and 0.9494. One exception was the farmland area (US-Arm site), where regional estimation and TM albedo verification results had $R^2$ lower than the other four surface types, and RMSE higher than other landforms. This may be due to the fact that farmland surface vegetation is more affected by anthropogenic factors than other land surface types; the jump in surface albedo makes the deviation of farmland albedo estimations larger than others. In this case, inducing more high-resolution images into the assimilation process could effectively improve the albedo estimation accuracy.

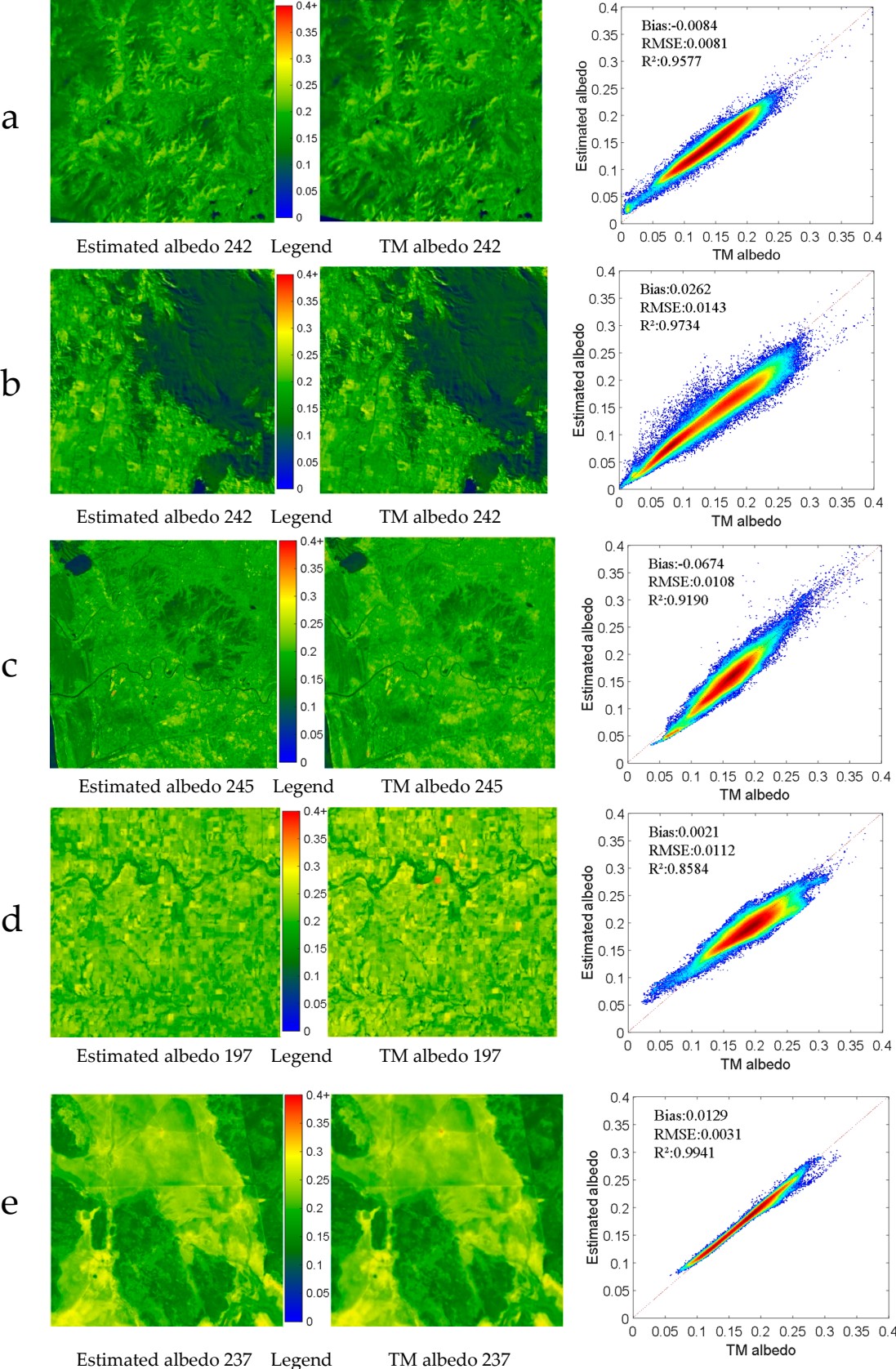

**Figure 8.** Shortwave albedo map generated by EnKF method with directly estimated TM albedo for five different land surface types: (**a**) DBF at US-Bar site (2009); (**b**) EBF at AU-Wac site (2007); (**c**) ENF at IT-Sro site (2009); (**d**) CRO at US-Arm site (2010); (**e**) GRA at AU-Stp site (2009). Right-hand scatter plots show consistency between the two sets of results, as number of dots gradually decreases from red to blue.

The DBF, EBF, and CRO sites selected in this study are heterogeneous research areas and have RMSE values less than 0.01 and $R^2$ values greater than 0.9, which indicate that the algorithm is well-suited to albedo estimation in heterogeneous areas.

## 5. Discussion

### 5.1. Accuracy of the Albedo Background

In the data assimilation algorithm, the background field is a preliminary estimate of target parameters, which is obtained from historical and empirical data, and reflects the general variation trend of the estimator [54]. Our EnKF algorithm results were obtained based on the optimal estimation of the current time. The background field trend introduces an observation increment in the optimal weight, so its accuracy is very important. We conducted spatial registration before constructing the background; after processing, the projection mode and resolution of MODIS data were consistent with Landsat. MODIS data actually provides background fields and dynamic models (preliminary estimates), so errors do exist, but they are allowed under the assimilation algorithm. Surface albedo presents a certain regularity in the growing season. We averaged the historical MODIS data, instead of the contemporary MODIS data, as the albedo background field because MODIS data were missing across our study area [55]. Consider the AU-DaP site as an example: The background field albedo has the same change trend as the original MODIS albedo, but with no missing data and an overall smoother structure. To this effect, the background field is more suitable for data assimilation.

The background field encompasses periodic albedo characteristics, but the albedo is overestimated or underestimated due to the accuracy of MODIS products (especially in heterogeneous pixels) [56]. Under the assimilation scheme, however, the background field only provides an initial prediction and the weight of background data is determined according to its accuracy. When the background accuracy is low, the weight is decreased and the weight of the TM observation is increased. Therefore, the estimation results gradually approach the field observations as the quantity of TM observations increases; using the mean albedo of the time series as a background field is feasible and effective.

### 5.2. Errors Induced by TM Albedo Estimation

In this study, TM albedos derived by direct estimation were utilized as albedo observation data. The TM albedos were used to adjust model-based predictions to bring them as close as possible to actual measured values. The quality of the TM data directly affects the accuracy of the final estimations. Previous researchers have thoroughly validated TM albedos derived by direct estimation [18]. In this study, we also compared the TM albedo with the ground measured albedo for several FluxNet sites. Figure 9 shows that the surface albedo estimated from TM images is highly accurate (maximum error < 0.03) and can be used as the input observation value for the data assimilation algorithm.

The accuracy of inversion is affected by the accuracy, quantity, and time distribution of TM data, where quantity and time distribution complement each other. In this study, we used L1T data with less than 10% cloud contamination for our estimations. There were approximately 5–11 TM data points available at the selected sites, which was sufficient for albedo estimation by data assimilation. If the quantity of available TM data in one year falls below 5 points, the albedo during the time series may not be updated for long periods of time, and the accuracy of the estimations degrades. Several high-quality observations are needed to ensure accurate estimations. Other high-resolution satellite images such as HJ1A/B or Sentinel-2 can be supplemented to resolve this problem for those areas where high-resolution observations are limited.

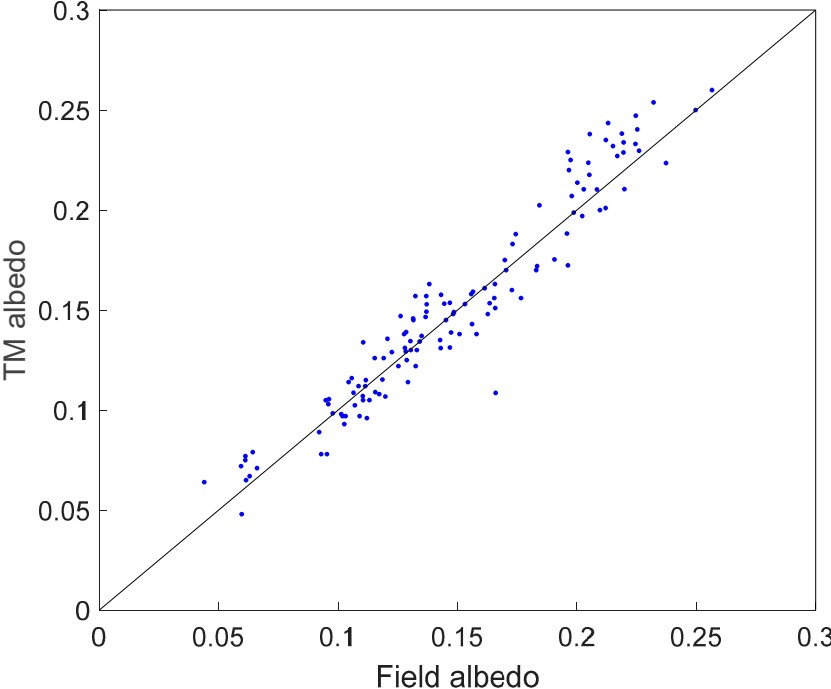

**Figure 9.** Directly estimated TM albedos and field albedos including all available TM images used in this study.

## 5.3. Error Setting in the Data Assimilation Algorithm

In the data assimilation system, the observation-model error is critical for land surface albedo estimation. We calculated observation error from field observations and TM data and calculated model error from field observations and the albedo background. Field observation data are key in terms of error determination. For each site, we used the field observation data to generate errors resulting in high estimation accuracy. When extending the model to regional use where no field observation data was available, we attempted to generate a "common" error by averaging the error of the 18 ground stations and inputting it to the assimilation scheme. The test results on the AU-DaP site are shown in Figure 10. Errors caused by TM data are relatively small and uniform, and using the mean errors instead of the original observation error did not affect the estimation results significantly. In effect, the common error from different sites represents the averaged field condition and can be used to account for large areas.

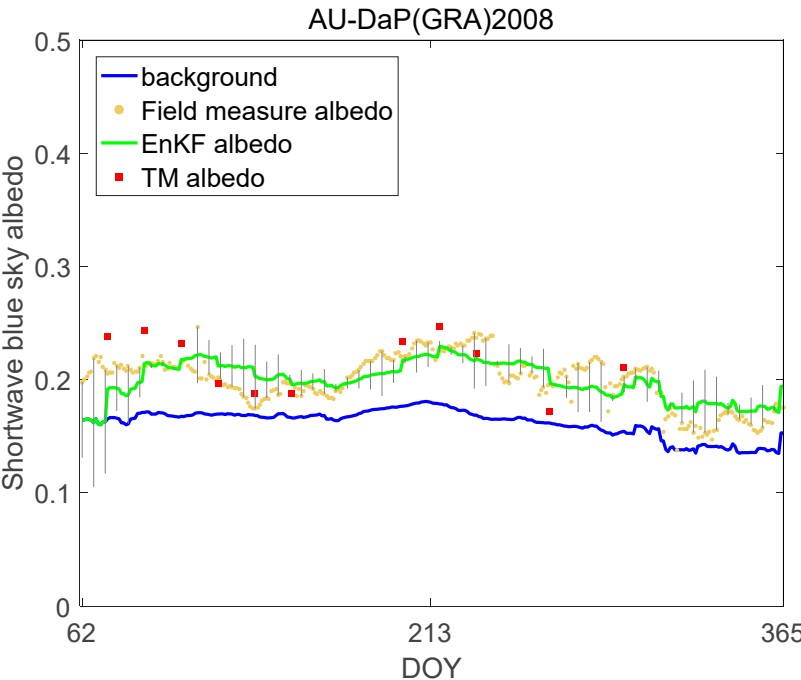

**Figure 10.** Estimated results using mean error of 18 stations as data assimilation input at AU-DaP. Replacing original observation error with mean error does not significantly affect the estimation results.

### 5.4. Capability of Capturing Abrupt Variations in Land Surface Albedo

Abrupt changes in surface albedo are often due to drastic changes in surface characteristics over a short period, which are difficult to capture using remote sensing data [57]. The background field as obtained from MODIS historical period data can reflect abrupt changes to a certain extent, but the effect is not obvious. If TM data are available during the period of abrupt change, the model prediction is updated and the estimated results are closer to the observed values; the abrupt change in surface albedo can be captured effectively.

In our time-series albedo estimation for farmland stations, surface albedo jump occurred on the 230th day at the DE-Kli site, the 150th day at the IT-Bci site, and the 230th and 250th days at the FR-Gri site. There are TM data available before and after these days, and the trajectory of the estimated results was corrected in time after adding the observed data. On the 170th day at the DE-Kli site, the 140th day at the IT-Bci site, and the 245th day at the FR-Gri site, there were no TM data available to update the model prediction and only the background field trajectory could be used for forward predictions. The abrupt variation in surface albedo is difficult to show and the estimated results contain deviations. The jump is more severe during snowy periods (especially snowfall and melting), where the surface albedo deviation may be higher than 0.5 and the EnKF algorithm cannot be applied effectively without support from TM data. To operate the surface albedo data assimilation algorithm, it is necessary to secure more high-quality satellite data as input observations. With support from more observations, the data assimilation algorithm is better able to capture abrupt variations and produce accurate estimations.

## 6. Conclusions

High spatio-temporal resolution albedo products are essential for climate simulations, as well as for various agricultural and environmental monitoring applications. In this study, we developed a novel method based on the ensemble Kalman filter algorithm to integrate MODIS high temporal resolution albedo and Landsat high spatial resolution albedo data to estimate high spatial-temporal resolution albedos.

We constructed an albedo background field using MODIS historical surface albedo data, from which the initial predicted albedo was obtained from a dynamic model. TM albedos derived by direct estimation were then used as observation data to update the initial predictions. The error of the dynamic model and observations was generated from field observations. When extending the model to regions where no field observation data were available, we created a "common" error by averaging the error of all test stations. The results were compared with ground measurement data for cropland, deciduous broadleaf forest, evergreen needleleaf forest, grassland, and evergreen broadleaf forest sites. We found that estimation accuracy is high (RMSE < 0.0152) for all land cover types. When applied to large areas, the proposed algorithm also shows high estimation accuracy both for homogeneous and heterogeneous regions. The estimated and TM albedos derived by direct estimation were in accordance with RMSE values of 0.003 to 0.0112, and $R^2$ values of 0.8584 to 0.9964.

The proposed algorithm has four distinct advantages over other similar methodologies. (1) It can generate reliable estimates of land surface albedo with high spatio-temporal resolution. (2) It can be used for all manner of snow-free vegetation surfaces, even heterogeneous surfaces. (3) It compensates for return-cycle and cloud contamination problems with high spatial resolution satellite image data. (4) The algorithm can be easily extended to other fine-resolution data similar to Landsat data (e.g., Sentinel-2).

The proposed algorithm does still have some drawbacks. For instance, the EnKF starts with the first available observation data to update the dynamic model, so the start time of high accuracy estimation depends on the availability of cloudless Landsat satellite images. The quality and quantity of observation data are crucial to the accuracy of the assimilation results. To further improve estimation accuracy, it is necessary to obtain more, higher-quality albedo data.

**Author Contributions:** Conceptualization, H.Z. and H.X.; Methodology, H.Z. and G.Z.; Software, H.Z. and G.Z.; Validation, G.Z.; Formal Analysis, H.Z. and G.Z.; Investigation, G.Z. and C.W.; Resources, H.Z., H.W. and J.W.; Data Curation, G.D.; Writing-Original Draft Preparation, G.Z.; Writing-Review & Editing, H.Z., J.W. and H.X.; Visualization, G.D. and C.W.; Supervision, H.Z. and H.X.; Project Administration, H.Z.; Funding Acquisition, H.Z.

**Funding:** This research was funded by the National Natural Science Foundation of China grant number 41801242, 41801366, the Key research and development program of China under grant 2016YFB0501502, the Chinese 973 Program under grant 2013CB733403, the Key Scientific and Technological Project of Henan Province under grant 172102110268.

**Conflicts of Interest:** The authors declare no conflict of interest.

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
