# Peer review of "Time Series High-Resolution Land Surface Albedo Estimation Based on the Ensemble Kalman Filter Algorithm"

_remotesensing, doi:10.3390/rs11070753_

Round 1

Reviewer 1 Report

This revision reflects significant efforts made by the authors to improve their manuscript based on this reviewer's comments. A few follow-up edits:

L72"...information types of varying resolution..."

L77: it's "Markov" not "markov"

L129: "...continuously imaged the land surface..."

L131: "...seven spectral bands sampled the shortwave range at a spatial resolution of..."

L135: "...can be obtained for individual pixels; TM data are prone..."

L414: Table 1 of the referenced paper by Sellers et al. lists requirements of +/- 0.02 in albedo, but you report an RMSE of < 0.05 as meeting those requirements. Please explain how you used the RMSE to estimate the uncertainty (i.e., error bars) for the albedo estimates in order to compare with the Sellers paper.

L419: replace "longer vacancy period" with "longer period of missing data" or similar

L515: "...10% cloud coverage..."

L555: might be better to use "discontinuity" (or perhaps "jump?") vs. "mutation" 

Author Response

Dear Reviewer:

Thank you for your comments! We have read all the comments and revised the paper carefully. The modifications are shown in blue font in the text. The main corrections in the paper and the responds to the reviewer’s comments are as following:

Comment 1: L72 "...information types of varying resolution..."

Response: Thanks for your comment! We have changed "... information of varying resolution..." to " ...information types of varying resolution... ". Please find the modification in line 72.

Comment 2: L77: it's "Markov" not "markov"

Response: Thanks for your comment! We have changed " markov " to " Markov ". Please find the modifications in line 78.

Comment 3: L129: "...continuously imaged the land surface..."

Response: Thanks for your comment! We have changed "continuously detected the land surface… " to " …continuously imaged the land surface… ". Please find the modifications in line 129.

Comment 4: L131: "...seven spectral bands sampled the shortwave range at a spatial resolution of..."

Response: Thanks for your comment! We have changed " …seven spectral bands covered the shortwave range at a resolution of… " to " …seven spectral bands sampled the shortwave range at a spatial resolution of… ". Please find the modifications in line 130.

Comment 5: L135: "...can be obtained for individual pixels; TM data are prone..."

Response: Thanks for your comment! We have changed " …can be obtained for a single pixel; TM data is prone… " to " …can be obtained for individual pixels; TM data are prone… ". Please find the modifications in line 135.

Comment 6: L414: Table 1 of the referenced paper by Sellers et al. lists requirements of +/- 0.02 in albedo, but you report an RMSE of < 0.05 as meeting those requirements. Please explain how you used the RMSE to estimate the uncertainty (i.e., error bars) for the albedo estimates in order to compare with the Sellers paper.

Response: Thanks for your comment! I am sorry for this mistake, "0.05" should be "0.02". This paper use the requirements listed by Sellers et al., and our albedo estimates (in Figure 6) met the 0.02 accuracy requirement in RMSE. We have changed "…The RMSEs for all land surface types are less than 0.05…" to "…The RMSEs for all land surface types are less than 0.02…", Please find the modifications in line 408.

Comment 7: L419: replace "longer vacancy period" with "longer period of missing data" or similar

Response: Thanks for your comment! We have changed " longer vacancy period " to " longer period of missing data ". Please find the modifications in line 414.

Comment 8: L515: "...10% cloud coverage..."

Response: Thanks for your comment! We have changed " …10% of cloud coverage… " to " …10% cloud contamination… ". Please find the modifications in line 521.

Comment 9: L555: might be better to use "discontinuity" (or perhaps "jump?") vs. "mutation"

Response: Thanks for your comment! We have changed " mutation " to " jump ". Please find the modifications in line 554,561.

Reviewer 2 Report

My comments from the previous review have been sufficiently addressed. 

Author Response

Dear Reviewer:

Thank you for your recognition of our work! Your approval is the biggest encouragement to us.

       With our best regars!

This manuscript is a resubmission of an earlier submission. The following is a list of the peer review reports and author responses from that submission.

Round 1

Reviewer 1 Report

The authors present an interesting method for fusing high spatial resolution satellite data with high temporal resolution data using an ensemble Kalman filter approach. It is a potential solution to a relevant problem faced by many who use satellite data products in their research. 

Below are some overall comments about the manuscript, followed by more specific comments and concerns; however, one general issue in particular merits additional emphasis: English usage. The authors' grammar and style are very inconsistent; in some places, the writing is already of publishable quality, and in others, it requires serious attention. In this reviewer's opinion, the article would benefit greatly if the authors were to enlist a copy editor who is proficient in English. To that end, I have not indicated below every instance where the article's English required revision; it is incumbent upon the authors to do so, and to edit their work accordingly.

General comments

- citation style: in lines 51-94 the citation style seems odd; please verify that this style is consistent with the journal's standards (I do not believe it is).

- abbreviation of ensemble Kalman filter: I believe the accepted abbreviation is EnKF, but the authors use more than one convention in the manuscript. Please be consistent.

- the abbreviation is properly spelled "FluxNet"; this needs to be changed throughout the paper

- Consider rearranging the presentation of figures and tables such that cover types are always in the same (consistent) order if possible; this would facilitate comparisons and general interpretability throughout the article.

Specific comments:

L2 (title): "...Based on the Ensemble..."

L18: "...high resolution observations are often unavailable..."

L28: "...less than 0.0152." Is there something significant about this particular value?

L31-33: remove the last sentence of the abstract

L47: "...mostly several times coarser than that of typical global object." Please explain what those are.

L75: "...sensitive parameters..." What are they?

L80: "...spatiotemporal integration..." What do you mean here?

L94: "...it is rarely applied in the estimation of surface albedo." Rarely, or never? Do you have any references you could cite here?

L104: change "vacancies" to "sparsity" or similar

L108-126: Use "upwelling" and "downwelling" as necessary (vs. "upstream", "downlink", etc.)

L131-144: This entire paragraph needs to be rewritten with proper, coherent English grammar. In particular, the sentence that begins at L137 with "Landsat TM..." seems important, but is nearly incomprehensible.

L150: change "one day" to "daily"

L153: Please explain why you used Ross Thick-Li Sparse instead of a different model

L159-160: use "kernel" vs. "nuclear"

L170: use "centered" vs. "focused"

L188: "...at the preceding time step..."

L204: "For the direct estimation approach..."

L279-280: "...analyze the effect of heterogeneity in the area around..."

L286-287: Reword the sentence beginning with "The range..." in order to describe more clearly the definition of range ("the high level distance" is not clear).

L303: "This provides a more detailed visual representation..."

Fig 3(a): Why do you not show the 2.0 km boundary, as given in Fig. 3(b)?

Fig. 3(b): What do the dashed lines represent? 

Table 2: The ranges given for IT-Col seem too low based on the plots in Fig. 3(b). Roughly, it appears that they should be around 200m at 1.0 km, and 275m at 1.5 km. Please explain this apparent discrepancy.

L335: "...five vegetated land surface..."

L344: What do you mean by "regularity?"

L346: What are the "accuracy advantages" of TM data in this context?

L347: What is meant by "the correct number of TM data?"

L361-364: "In addition, the variation of surface albedo for the snow cover period fluctuates greatly, and the data for MODIS albedo do not reflect the changing characteristics of surface albedo during the snow cover period. Therefore, this algorithm is not suitable for estimation of surface albedo during snow cover periods." Why not? Please explain this claim as it is a serious limitation of the approach presented in this paper.

L365: "...five surface types were generated based..."

L379: "...has been considered to meet the requirements of global and regional climate models" Please provide at least one independent citation that supports this claim.

L381-383: Please explain why ENF and CRO show the highest accuracies. Are these differences between cover types to be expected?

L386: "...when it has already met the requirements." What does this mean?

L460: "We do not use MODIS data as the background field..." Do you mean here something like "native" or "contemporary" MODIS data (vs. your "climatology" method)?

L464: "...but the background field has no missing data and looks smoother."

L487: The first sentence of this paragraph needs to be rewritten using coherent English grammar.

L489-492: Please explain how you determined the numbers of TM images required for "accurate" assimilation results.

L503-509: Have you considered developing separate "common" error sets for all of the sites within each cover type, rather than a single, global error set across all sites and cover types?

L533: "...agricultural and environmental monitoring..."

L557-561: Please add to the listed drawbacks that the authors' approach is inappropriate for areas having frequent or persistent snow cover.

Reviewer 2 Report

This paper used the Ensemble Kalman Filter Algorithm to estimate 30m albedo based on the inputs of 500m MODIS albedo product and the calculated 30m Landsat albedo. This may make sense, but the input of the calculated 30m Landsat albedo using empirical linear relationship is inaccurate. Kalman Filter is a recursive algorithm, and the estimated values in each step can be regarded as a weighted average between the model value and observed value. If the inputs are inaccurate, the final results cannot be reliable. Second, it can't achieve the goal of high spatial and temporal resolution, and it can only achieve the goal of high spatial resolution. Without a reliable 30 m Landsat albedo input, there is no reliable final estimate, let alone an increase in temporal resolution. Therefore, I recommend rejecting this paper for publishing in Remote Sensing.

Some other comments:

First, the structure of this paper is very confusing.

1.       For example, there are two equations in the results (chapter 4.1). All descriptions should be written in the Methods part.

2.       No descriptions are given for the study area. There should be a chapter named “2.1 Study area”. And “2. Experimental data” should be “2. Study area and data”.

3.       In the methods, there should be a paragraph which summarized the whole methods before chapter 3.1, instead of directly giving three chapters.

4.       There should be an extra chapter named “evaluation”, “assessment” or “validation” in the methods. In this chapter, the authors should give the details of validation and indicate the metrics for the accuracy.

Second, the manuscript is poorly written, there are many mistakes about the RS required formats such as the citation and the font and font size.

5.       L7: RS requires the emails for each author.

6.       L28: RMSE. The full name of each abbreviation should be given when it first appears.

7.       There should be a space between the word and the citation number.

8.       L54: “Shuai et al. Proposed an algorithm for generating land surface albedo at 30 m resolution 55 using Landsat and the anisotropy information of MODIS observations[17].” “Shuai et al. XXX[17]” should be “Shuai et al. [17] XXX.” “Proposed” should be “proposed”. Same mistakes should all be modified.

9.       The logic of introduction is very poor.

10.   L225: change the fond of “the terms”.

11.   L286: (Cooper et al., 1997) should use the citation style of RS.

12.   Change the font size of words such as “R^2 > 0.6”, “R^2” and so on.

13.   L436: “R2” should be “R^2”.

Reviewer 3 Report

Please see attached review.
